# Coherent diffractive imaging of microtubules using an X-ray laser

Gisela Brändén[1], Greger Hammarin [1], Rajiv Harimoorthy[1], Alexander Johansson[1], David Arnlund[1],
Erik Malmerberg[2], Anton Barty[3], Stefan Tångefjord[1], Peter Berntsen[1], Daniel P. DePonte[4], Carolin Seuring [3,5],
Thomas A. White[3], Francesco Stellato[3], Richard Bean[3], Kenneth R. Beyerlein [3], Leonard M.G. Chavas[3],
Holger Fleckenstein[3], Cornelius Gati[3], Umesh Ghoshdastider[6], Lars Gumprecht[3], Dominik Oberthür[3],
David Popp[6], Marvin Seibert[4], Thomas Tilp[3], Marc Messerschmidt[4], Garth J. Williams[4], N. Duane Loh[7],
Henry N. Chapman [3,5,8], Peter Zwart[2], Mengning Liang[3,4], Sébastien Boutet[4], Robert C. Robinson[6,9,10] &
Richard Neutze [1]

X-ray free electron lasers (XFELs) create new possibilities for structural studies of biological objects that extend beyond what is possible with synchrotron radiation. Serial femtosecond crystallography has allowed high-resolution structures to be determined from micro-meter sized crystals, whereas single particle coherent X-ray imaging requires development to extend the resolution beyond a few tens of nanometers. Here we describe an intermediate approach: the XFEL imaging of biological assemblies with helical symmetry. We collected X-ray scattering images from samples of microtubules injected across an XFEL beam using a liquid microjet, sorted these images into class averages, merged these data into a diffraction pattern extending to 2 nm resolution, and reconstructed these data into a projection image of the microtubule. Details such as the 4 nm tubulin monomer became visible in this reconstruction. These results illustrate the potential of single-molecule X-ray imaging of biological assembles with helical symmetry at room temperature.

[1] Department of Chemistry and Molecular Biology, University of Gothenburg, Box 462, 40530 Gothenburg, Sweden. [2] Molecular Biophysics and Integrated Bio-Imaging Division, Lawrence Berkeley National Laboratory, 1 Cyclotron Rd, 94720 Berkeley, CA, USA. [3] Center for Free-Electron Laser Science, Deutsches Elektronen-Synchrotron DESY, 22607 Hamburg, Germany. [4] Linac Coherent Light Source, SLAC National Accelerator Laboratory, Menlo Park, CA, USA. [5] The Hamburg Center for Ultrafast Imaging, 22761 Hamburg, Germany. [6] Institute of Molecular and Cell Biology, Biopolis, A*STAR (Agency for Science, Technology and Research), 138673 Singapore, Singapore. [7] Department of Physics, National University of Singapore, 117551 Singapore, Singapore. [8] Department of Physics, University of Hamburg, 22761 Hamburg, Germany. [9] Department of Biochemistry, National University of Singapore, 117597 Singapore, Singapore. [10] Research Institute for Interdisciplinary Science, Okayama University, Okayama 700-8530, Japan. Correspondence and requests for materials should be addressed to G.B. (email: gisela.branden@gu.se) or to R.N. (email: richard.neutze@gu.se)

Biological filaments are versatile components of all cells. Functional filaments are found within the cytoskeleton, in muscle fibers and constitute the bacterial flagella. Some filaments such as amyloid fibrils are involved in several neurodegenerative diseases such as Alzheimer's and Parkinson's diseases. One member of the family of filamentous assemblies is the microtubule, which is a major constituent of the cell cytoskeleton and it gives structure to the cell and participates in vital processes such as intracellular transport and chromosome segregation during cell division[1–3]. Microtubules are built up of α/β tubulin dimers arranged as hollow cylinders with a mean diameter of ~24 nm and consist of 12 to 16 multiples of tubulin dimers per turn, the so-called proto-filaments, assembled in a staggered conformation with most helices arranged with a pitch of three tubulin monomers per turn. The dominant variant in vivo is the 13-protofilament (13-pf) form[4]. Microtubule polymerization grows through the addition of GTP-bound tubulin dimers. After polymerization, GTP is hydrolyzed to GDP whereupon the GDP-bound tubulin dimer can depolymerize from the microtubule, giving rise to what is termed dynamic instability[5] during which microtubules rapidly switch between cycles of growth and shrinkage within the cell.

X-ray crystallography has revolutionized the biological sciences by providing atomic-level information for a large variety of components of living cells, leading to a detailed understanding of complex biological phenomena as diverse as the harvesting of solar energy to the synthesis of proteins. The major limitation with X-ray crystallography, however, is that the method can only be used to study biological samples that form well-ordered crystals. Although for many biological targets this restriction is not very limiting in practice, for almost all classes of biological filaments, diffraction quality crystals are difficult to grow. Our current knowledge of the structure of microtubules therefore stems primarily from negative-stained electron microscopy (resolution down to ~3 nm[6]), X-ray fiber diffraction (resolution ~1 nm[7]), X-ray solution scattering (resolution ~3 nm[8,9]) and cryo-electron microscopy studies[10–15] which have achieved a resolution (~3.5 Å) at which most side-chains can now be resolved. Studies of microtubule polymers are complemented by high-resolution structures of the α/β heterodimer solved using electron or X-ray diffraction[16–19]. Despite the impressive progress of single-particle cryo-electron microscopy in particular, that method uses flash-frozen samples and the technique is therefore limited to the investigation of static structures frozen to cryogenic temperatures. X-ray methods that enable data to be collected at room temperature have the potential to allow the study of the dynamics of microtubule growth and shrinkage and may complement higher-resolution cryo-electron microscopy methods.

X-ray free electron lasers (XFELs) offer new possibilities to advance the state-of-the-art for studying biological filaments. The key idea is known as diffraction before destruction and suggests that by using ultrafast highly intense X-ray pulses it is possible to outrun X-ray induced damage and thereby collect diffraction data before the sample is destroyed[20]. This concept was first validated using micro-fabricated 2D samples at a soft X-ray laser[21] and now provides the basis for serial femtosecond crystallography of micro-crystalline protein samples to high-resolution[22,23]. Coherent X-ray diffraction image reconstructions from data collected at XFELs have been reported for viruses in projection[24,25] and in 3D[26,27] as well as for carboxysomes[28] and mitochondria[29]. Despite these advances, the resolution of the image reconstructions is limited to a few tens of nanometers[30]. Biological polymers that possess helical symmetry represent an intermediate step between 3D crystals and single molecules due to their 1D translational symmetry along the helical axis. To this end, it was recently demonstrated that F-actin microfilaments,

amyloid fibrils and pili flow align when injected within a microjet across a focused XFEL beam[31], with the order of 100 polymers per sample being aligned to within 5°. Similar studies examined and sorted X-ray scattering from crystalline amyloid fibrils[32] and reported diffraction imaging of aligned amyloid fibrils held upon a graphene support[33]. These approaches therefore offer a promising path towards extending the resolution of single biomolecule imaging at an XFEL.

Here we collected X-ray diffraction data from microtubules injected across an XFEL beam using a gas dynamic virtual nozzle liquid microjet[34,35]. Given the protein concentration, microjet diameter and the XFEL beam focus, ~20 microtubules were sampled within the exposed volume for every XFEL exposure. We sorted these images into class averages using software adapted from single-particle cryo-electron microscopy applications[36,37]. Data were merged to recover a single 2D diffraction pattern extending to 2 nm resolution from which we made a 2D projection image reconstruction using iterative phase retrieval assuming that the density outside of the microtubule was zero[24,38,39]. This analysis recovered the characteristic diameter of the microtubule distributions and resolved 4 nm sub-structure corresponding to the individual tubulin monomers that was not included in the initial phases. As such we demonstrate a simple approach for recovering a 2D projection image of a biological polymer from XFEL data by applying tools developed for the analysis of single-particle electron microscopy data and a phase retrieval algorithm. Future advances in XFEL intensity and focus, as well as improving sample handling and injection procedures, may allow dynamical processes to be imaged at room temperature to high resolution.

## Results

**Data collection using XFEL radiation.** Polymerized samples of bovine microtubules were pre-formed and stabilized using the anti-cancer agent taxol[40]. The characteristics and integrity of the sample were confirmed by small-angle X-ray scattering (Supplementary Fig. 1a) and negative-stain electron microscopy (Supplementary Fig. 1b). Microtubule suspensions were injected in vacuum across a highly focused (~200 nm focal diameter) XFEL beam using a gas dynamic virtual nozzle[34,35] at the Coherent X-ray Imaging (CXI) experimental station[41] of the Linac Coherent Light Source (LCLS)[42] at the SLAC National Accelerator Laboratory. Each 6 keV X-ray exposure contained ~2 mJ of energy (~$2 \times 10^{12}$ X-ray photons per pulse) and was ~33 fs in duration. X-ray scattering images were read out on a CSPAD X-ray detector[43] at a rate of 120 Hz which matched the XFEL repetition rate (Fig. 1).

Background scattering was minimized by aligning the XFEL beam to intercept the microjet at its minimum diameter of approximately 5 µm but prior to the microjet breaking into droplets. The protein concentration of 5 mg/ml tubulin that was used throughout these studies meant that ~20 microtubules were contained within the ~0.16 µm³ ($\pi \times$ (200 nm/2)² $\times$ 5 µm) volume intercepted by the XFEL beam. An added advantage of the microjet was that microtubule polymers flow aligned to a very high degree[31] as the jet was focused using the gas dynamic virtual nozzle. Thus despite the fact that the X-ray scattering recorded from a single XFEL pulse derived from only approximately 20 polymers, this alignment meant that individual detector images showed diffraction spots along an equatorial line that are characteristic of microtubule fiber diffraction (Fig. 2a). In order to produce a submicron diameter jet with fewer microtubules within the sampled volume the sample needed to pass through a thinner capillary. This was difficult to maintain due to the viscous nature of the microtubule slurry and this tended to block the microjet. Moreover, even when a flow could be maintained, a sub-

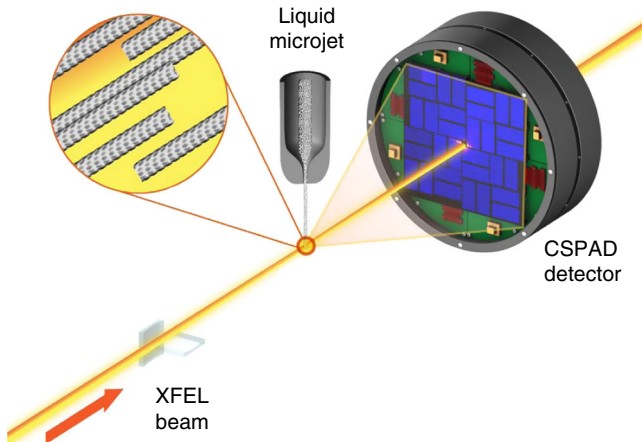

**Fig. 1** Schematic illustration of the experimental setup. Samples of pre-formed microtubules were injected across a focused X-ray free electron laser (XFEL) beam. Diffraction data were recorded on a CSPAD X-ray detector which was read out at 120 Hz, which matched the incoming repetition rate of the XFEL. Under the experimental conditions approximately twenty microtubules were intercepted by the X-ray beam as it passed through the microjet. Figures 1, 2, and 5 were originally presented in the doctoral thesis of Harimoorthy[59]

micron jet was not optimal for this application since its flow was unstable and therefore the probability that the XFEL beam intercepted the edge of a microjet was high. The resulting edge diffraction pattern was very strong and dominated over the desired measurement of microtubule X-ray scattering.

**Data processing and sorting.** Almost one million images were collected in a single experimental shift. After pre-processing with the Cheetah software package[44] ~10% of these images were selected for further analysis based upon manual inspection of representative images from different collection runs (Fig. 3). These images were then sorted based upon general statistics to separate microtubule diffraction patterns from background solution scattering or scattering from the edge of the jet stream[36] after which ~60% of the images were discarded (Fig. 3). The remaining 38,588 diffraction patterns were further classified using the automated algorithm Xmipp[36,37]. Class averaging was performed on blocks of ~1000 images that were divided into five subgroups each using a self-organizing maps (SOMs) neural network algorithm[37]. This classification step was dominated by two main factors: the quality of X-ray diffraction from the microtubules, since there is variation in signal to noise and background between images; and the instantaneous orientation of the microjet at the moment of exposure to the XFEL beam, which is reflected in the

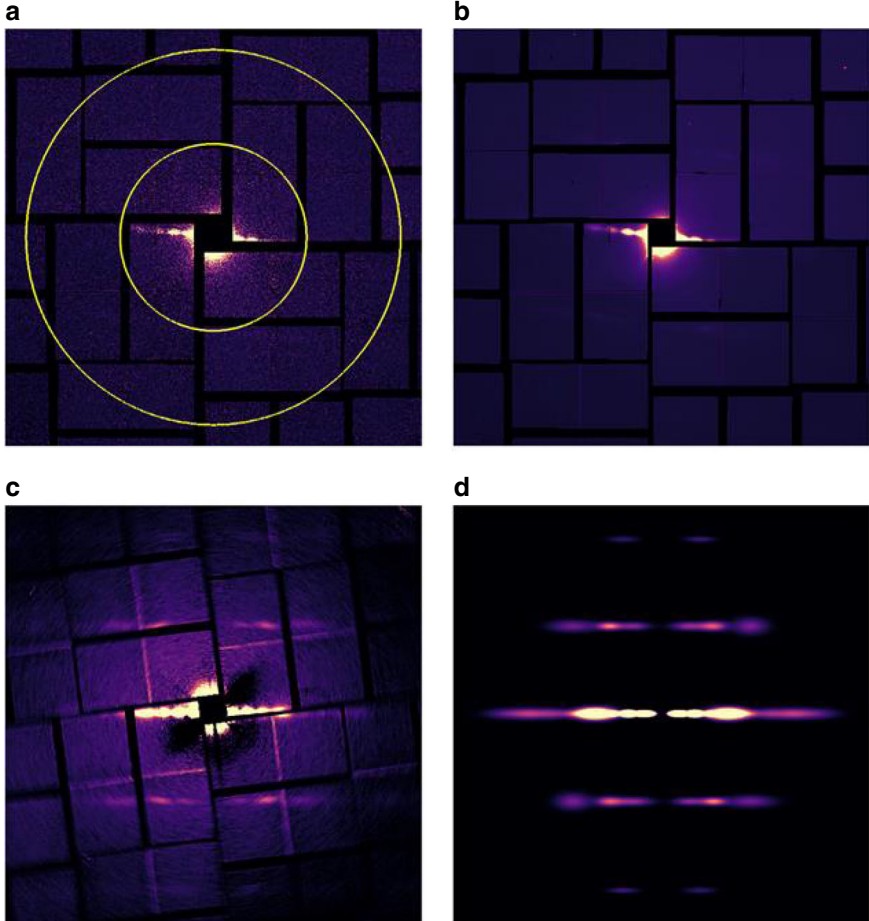

**Fig. 2** X-ray diffraction images and class averages. **a** X-ray diffraction image recorded from exposure of microtubule samples to a single XFEL pulse. The inner yellow circle indicates 4 nm resolution and the outer circle indicates 2 nm resolution. This scale applies to all panels. **b** Average of approximately 200 images selected by class average sorting of the X-ray diffraction patterns using software originally developed for electron microscopy applications[36]. **c** Sum of the class averages (13,511 images summed in total) after aligning each class average by rotating about the beam center. **d** Fitted diffraction image recovered by fitting Gaussian peaks to the features identified as diffraction peaks in (**c**)

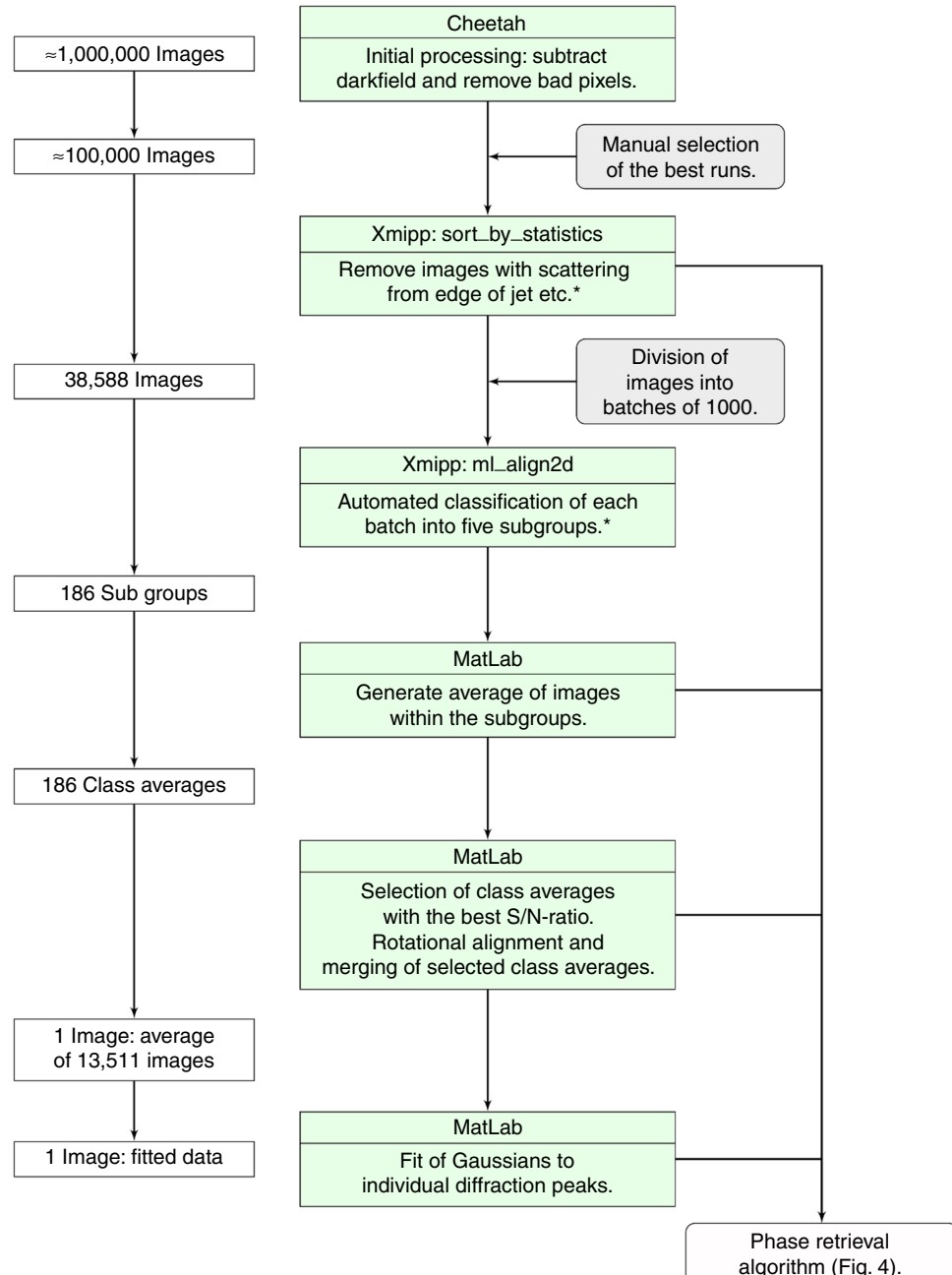

**Fig. 3** Flow chart for data processing of the diffraction images. Manual intervention is indicated in gray boxes. An asterisk indicates where data processing steps used only the four central detector panels

angle of the equatorial diffraction line relative to the X-ray detector geometry. These steps therefore corrected for fluctuations in the direction of the microjet during each experimental run. A class-average diffraction pattern for each of the 186 subgroups was recovered by summing all diffraction images from within a single class (Fig. 3). Most class averages displayed diffraction along the so-called layer line at 4 nm resolution (Fig. 2b) which corresponds to the longitudinal distance between the tubulin monomers that make up the microtubule. From these 186 class averages, 69 were selected based on a histogram analysis of the contrast of the diffraction spots along the 4 nm layer line, with those having the strongest signal-to-noise being kept. The orientation of the equatorial diffraction line was then determined and class averaged images were rotated prior to being super-imposed and further averaged (Fig. 3). The result was an

optimized diffraction image which consisted of data merged from a total of 13,511 individual X-ray exposures and the higher resolution 2 nm layer line also became visible (Fig. 2c).

Because fibril diffraction has 1D translational symmetry, the X-ray diffraction image recorded from aligned samples shows intense peaks analogous to the diffraction peaks visible in protein crystallography. We therefore adopted the philosophy used in X-ray crystallography and fitted the individual diffraction peaks with horizontal and vertical Gaussian functions as a final step of data optimization (Fig. 2d). This final step (Fig. 3) removed all sources of noise between the diffraction spots and therefore spurious features in these regions such as background scattering from the solution, variations in the background of detector panels and bad pixels, as well as artefacts from the gaps between the detector tiles, did not hinder image reconstruction.

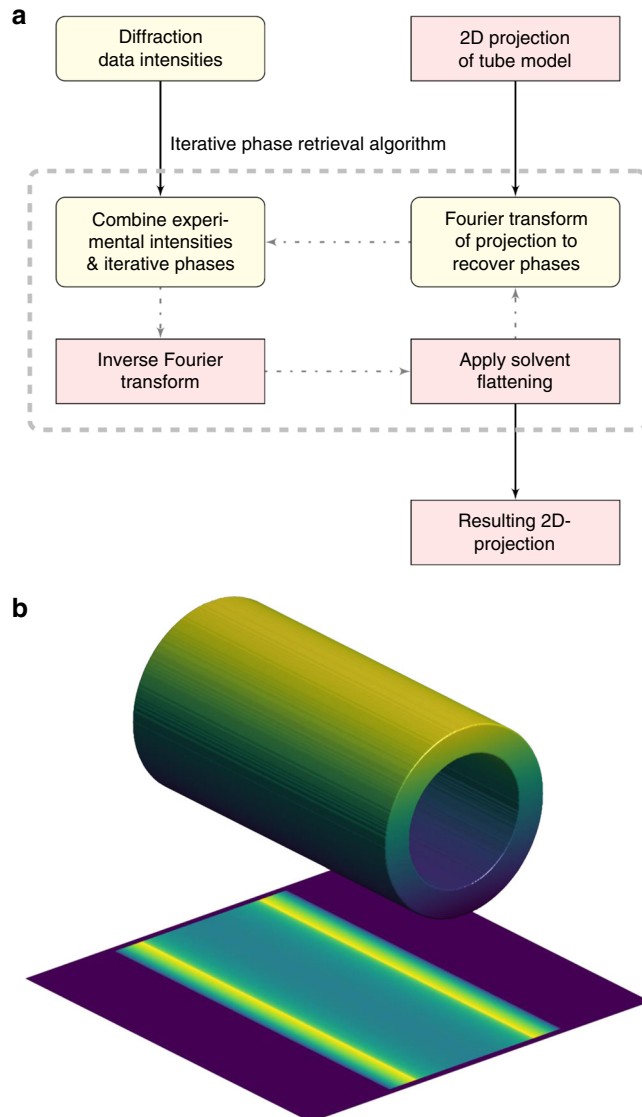

**Fig. 4** Iterative phase retrieval algorithm. **a** Flow chart illustrating the input of X-ray diffraction data, initial phases, application of solvent flattening (support function) and forward and backward Fourier transformations. Pink boxes represent real space and cream boxes represent reciprocal space. **b** Initial phases were provided from a projection of a featureless tube onto a 2D image

**Iterative phase retrieval**. A simple iterative phase retrieval algorithm (Fig. 4a) was implemented in MatLab to invert the diffraction images shown in Fig. 2. This recovered 2D projection images of the sample based upon the principle that objects are oversampled in coherent diffractive imaging and therefore phases can be improved iteratively if one assumes that the electron density is zero outside of the object[24,38,39]. Microtubules are smaller than the diameter of the focused XFEL beam in one direction but extend further than this diameter in the perpendicular direction. We therefore created a support outside of the microtubule by forcing the projection density of the object to zero for $|x| \geq 15.6$ nm (falling as a half Gaussian), which corresponds to the horizontal direction relative to the image orientation in Fig. 5. This step is analogous to solvent flattening in X-ray crystallography. We also created a support in the perpendicular direction by modulating the density of the object by a Gaussian in the $y$-direction (full width half maximum of 159 nm) to ensure that the projected density also fell to zero in the vertical direction. This perpendicular Gaussian modulation was inverted after the

phase retrieval algorithm had converged. A 2D projection of a featureless tube of 25.4 nm outer diameter and 17.4 nm inner diameter was used for initial phases (Fig. 4b). Because it is not physically possible to record the direct forward X-ray scattering (the [0,0] peak in X-ray crystallography) we also used this featureless tube to predict this missing data during iterative phase retrieval, where we scaled the height of the central peak so that its adjacent theoretical peak matched the amplitude of the corresponding experimental peak along the equatorial line.

Figure 5 presents the results of applying this Fourier inversion algorithm to recover 2D projection structures from each of the images shown in Fig. 2. It is apparent that even a single shot (Fig. 2a) contains some structural information concerning the microtubule diameter (Fig. 5a) and that the information content improves after each step of image processing. When all class-averages are aligned and merged together into a single image (Fig. 2c) the borders of the microtubule become clearly defined after Fourier inversion, with a peak-to-peak diameter of 20 nm (Fig. 5c). This distance corresponds to the maximum density of the microtubule wall in projection and is therefore lower than the outer diameter distance of ~25 nm that is usually quoted for microtubules. When the idealized noise-free X-ray image (Fig. 2d) that was reconstructed from Gaussian fits to the X-ray diffraction spots visible in Fig. 2c is Fourier inverted, more structural details emerged. In particular, both the sharpness of the microtubule boundaries are improved and substructures 4 nm in length become visible from the image reconstruction (Fig. 5d). This length-scale is equal to the length and diameter of a tubulin monomer and we therefore conclude that, despite the limitations of averaging over non-aligned samples, these steps of image processing lead to additional structural information that is highly relevant biologically.

In vitro preparations of microtubules yield varying mixtures of 11-pf, 12-pf, 13-pf, 14-pf, 15-pf, and 16-pf polymers[45–47] (i.e., microtubule helices containing 11 to 16 tubulin dimers per turn) depending upon the details of the preparations. To test the sensitivity of our iterative phase retrieval algorithm to the parameters of the featureless tube starting model (Fig. 4b) we repeated the procedure above using a variety of starting inner and outer diameters with a center of mass (COM) radius varying from 9 nm to 16 nm. Our results showed that, when using starting COM radii from 10 nm to 15 nm, all iterative phase retrieval results converged to a peak-to-peak separation of 19.9 ± 0.8 nm (Supplementary Fig. 2a). Moreover, a plot of the Pearson correlation function for the X-ray scattering intensities recovered by a Fourier Transform of the final 2D reconstruction calculated against the low-resolution experimental peaks along the equatorial line of Fig. 2c (center left inner panel) showed a fairly broad distribution peaked about a COM radius of 11 nm (Supplementary Fig. 2b). If one approximates a cryo-electron microscopy structure of a 13-3 pf biological assembly[13] (pdb entry 5SYF) as a featureless tube, this yields a center-of-mass radius of 11.2 nm, which is in good agreement with these findings. Alternatively, azimuthal integration of the image shown in Fig. 2c generates a pseudo SAXS image with the $J_{01}$ peak at 0.297 nm$^{-1}$. Using the formula[48] $J_{01} = 7.66/2\,R$ this predicts a mean helical radius of 12.9 nm (as distinct from the COM radius above), which corresponds to an average proto-filament number[48] of 14. Thus the image reconstruction algorithm is robust and is consistent with a microtubule population being dominated by a mixture of 13-pf and 14-pf microtubule forms.

## Discussion
Microtubule dynamic instability, the stochastic switching between phases of growth and shrinkage that is essential for microtubule

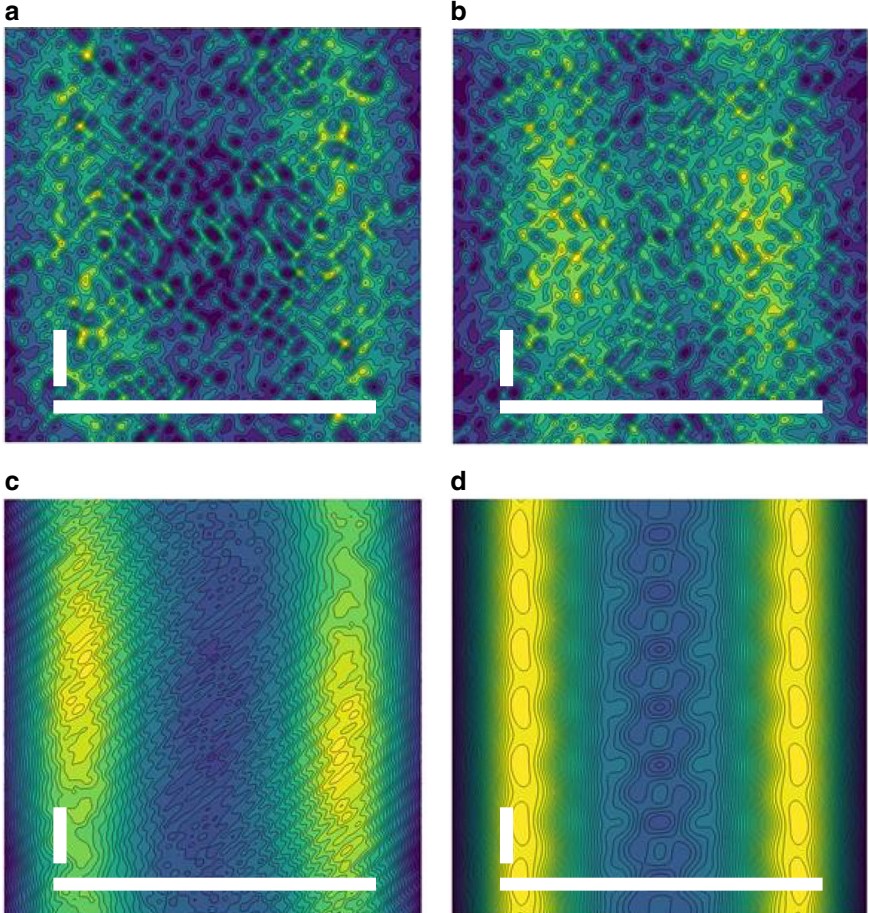

**Fig. 5** Projection images recovered by iterative phase retrieval. **a** 2D projection image of microtubule samples recovered after Fourier transform of the image in Fig. 2a using a featureless tube for initial phase and after 100 cycles of iterative phase retrieval. **b** 2D projection image recovered after 100 cycles of iterative phase retrieval of the image shown in Fig. 2b. **c** 2D projection image recovered after 100 cycles of iterative phase retrieval of the image shown in Fig 2c. **d** 2D projection image recovered after 100 cycles of iterative phase retrieval of the image shown in Fig. 2d. These projection images shows increasing detail after each processing step. Both the average microtubule diameter of approximately 25 nm and sub-structures of 4 nm become visible when inverting the fitted diffraction image. White bars indicate 25 nm in the horizontal direction and 4 nm in the vertical direction. Blue represents low projection density whereas yellow is high projection density

function, was introduced conceptually thirty years ago[5] but is still not very well understood on a molecular level. One striking example occurs during mitosis when chromosome segregation is driven by microtubule depolymerisation[1–3]. To describe in detail these phenomena will require novel approaches to structural analysis that capture how the different structural states interconvert, how they influence polymerization dynamics, and how this is related to the chemical kinetics of GTP hydrolysis. Although understanding of microtubule structure has advanced significantly over the last two decades and the recent achievements of cryo-electron microscopy studies of microtubules are deeply impressive[12,49], it will be necessary to study the structure of microtubules at physiological temperature and in solution if we are to address these questions.

We used XFEL radiation to probe the average room-temperature structure in projection of a slurry of microtubules that are largely aligned within a microjet. This extends earlier reports of X-ray diffractive imaging of other filamentous systems[31–33] by using imaging sorting techniques pioneered for electron microscopy to sort and average images[36], and by applying a simple iterative phase retrieval to recover 2D projection images of microtubules. This approach allowed data to 2 nm resolution (Fig. 2c, d) to be incorporated into the reconstruction from which structural details became visible with a characteristic

length scale of 4 nm (Fig. 5d). In this manner the amplification of the diffraction signal due to the presence of 1D translational symmetry has facilitated a significant advance in resolution over earlier coherent diffractive imaging studies of virus particles using XFEL radiation[24,26] that achieved a resolution of 32 nm in projection[24] and 125 nm[26] and 28 nm[27] after 3D reconstruction, or projection images of live cyanobacteria[50] and carboxysome[28] that were recovered to a resolution of 75 nm and 18 nm respectively.

A limitation of the data we present is that each useful image contained diffraction from ~20 microtubules within the X-ray exposed volume. It is anticipated that XFEL data collection will be improved through technical advances such as a tighter XFEL focus, more photons within each XFEL pulse and improved detector sensitivity and stability[30]. By combining these advances with developments leading to more stable submicron jet delivery systems[35] or fixed-target sample manipulation[33,51] we believe that interpretable single-shot X-ray diffraction data from individual microtubules may soon be within reach. Sample delivery systems may also be mounted upon rotation stages, which will facilitate the collection of coherent diffractive images of microtubules from multiple angles and thereby better sample reciprocal space. As illustrated from theoretical reconstructions using a simple helix model of a microtubule (Fig. 6), if single snapshots can be recorded from biological filaments and sorted into different class

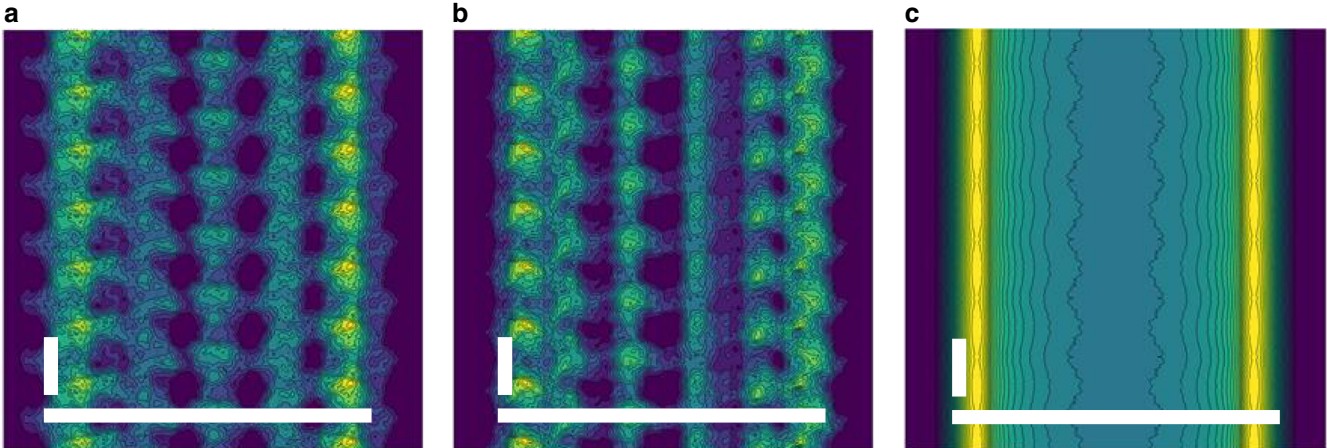

**Fig. 6** Oversimplified model illustrating single fibril snapshots. **a** Iterative phase retrieval image recovered from theoretical data generated from projection images of a single snapshot of a simplified helical model constructed from 4 nm spheres. **b** Iterative phase retrieval image recovered from the same object but rotated by 66° relative to its orientation in (**a**). **c** Iterative phase retrieval image recovered from the same object but averaged over 360° of rotation about its helical axis. White bars indicate 25 nm in the horizontal direction and 4 nm in the vertical direction

averages, then projection images from individual particles (Fig. 6a, b) have the potential to yield considerably more structural information than when each projection image is first merged together (Fig. 6c).

Single particle electron microscopy took decades to develop[52] but has ushered in a revolution in structural biology[53] as high resolution cryo-EM structures from challenging biological systems that were not amenable to crystallization have emerged. Foreseeable advances in the coherent diffractive imaging using XFEL radiation suggest that it will become possible to classify diffraction images according to the orientation of individual microtubules[32]. It may also become possible to further sort room-temperature diffraction data according to whether the microtubule is probed along its helical structure or at either end of the microtubule, where the association and dissociation of tubulin dimers[54] that underlies the biologically critical mechanism of dynamic instability is ongoing at room temperature. Such advances would open up unique possibilities for studying the dynamics of microtubule formation and dissociation in a controlled manner, potentially yielding biological insights that cannot be accessed with cryogenic approaches.

## Methods

**Protein preparation**. Tubulin from calf brains was purified by two cycles of polymerization/depolymerisation in the presence of a high-molarity PIPES buffer as previously described[55], which also successfully removes microtubule associated proteins. The assembly of purified tubulin into microtubules was monitored by a turbidimetry assay[56] to confirm functionality. Tubulin at a concentration of 45–180 µM in 80 mM PIPES, 2 mM $MgCl_2$, 0.5 mM EGTA, 10% glycerol pH 6.9 was mixed with 2 µM GTP and polymerization was measured spectro-photometrically at 37 degrees as an increase of the absorbance at 350 nm.

**Microtubule formation and characterization**. Microtubules were formed by incubating purified α/β-tubulin at 90 µM (corresponding to 10 mg/ml) in 80 mM PIPES, 2 mM $MgCl_2$, 0.5 mM EGTA pH 6.9 with 2 µM GTP at 37 degrees. Pre-formed microtubules were stabilized by the addition of 10 µM taxol. The structural integrity of the microtubules was confirmed by small-angle X-ray scattering collected at 17 degrees, to mimic the environment of the experimental hutch at LCLS, at the I911-4 beamline of MAX-lab, Sweden. Microtubules were visualized using negative-stain electron microscopy after passing them through a filter with a pore size of 20 µm under high pressure, which mimicked the filtering step before injection of the sample at the LCLS. This analysis confirmed that microtubules remained intact.

**XFEL data collection**. XFEL diffraction data were collected at the CXI instrument at the LCLS. The pre-formed taxol-stabilized microtubules were diluted twice (corresponding to a tubulin dimer concentration of 45 µM or 5 mg/ml) in 80 mM

PIPES, 2 mM $MgCl_2$, 0.5 mM EGTA pH 6.9 and injected using a liquid microjet formed with a Gas Dynamic Virtual Nozzle[34,35] at a velocity of approximately 10 ms⁻¹, into a vacuum chamber where they intersected with the highly brilliant X-ray pulses of ~33 fs duration and 6 keV energy (wavelength = 2.07 Å) focused to a spot size of 0.04 µm². The XFEL repetition rate and the detector readout rate were both at 120 Hz and the sample-to-detector distance was 565 mm. Diffraction data were recorded on a Cornell-SLAC pixel array detector (CSPAD)[43].

**Data processing**. A flow chart describing the data processing steps is shown in Fig. 3. Initial processing of almost one million diffraction images was done using the Cheetah software[44], where darkfield frames were subtracted and unreliable pixels removed. Selected images from different collection runs were manually inspected and the best runs, based upon resolution and angle of incidence of the jet stream, were selected for further processing. Around 100,000 images were then sorted based on general statistics to identify outliers using the program image_-sort_by_statistics from the imaging processing software Xmipp[36,37]. This step enabled us to get rid of images where scattering from the edge of the jet stream or background scattering from the solvent dominated over the diffraction from microtubules. A conservative cut-off was applied where the highest scoring 40% of the sorted images were selected for further processing. The remaining 38,588 images were divided into groups of about 1000, where each group was classified into five automatically defined subgroups with the program ml_align2d of Xmipp according to angle of the equatorial diffraction line relative to the X-ray detector geometry using a maximum likelihood approach[37,57]. The number of subgroups for each sub-set of 1000 images was chosen to be five in order to facilitate the analysis using relatively modest computing capabilities since these calculations were computationally expensive. These sorting and classification steps were made faster by using only the central four out of sixteen detector panels of each image. An average diffraction pattern (Fig. 2b) was recovered for each of the 186 subgroups that resulted from this step (Fig. 3) by summing all diffraction images from within each subgroup. An intensity histogram along the 4 nm layer line in the average diffraction pattern of each subgroup was calculated. The intensity of the two most distinct peaks was compared to the intensity across the entire line and the average diffraction patterns with highest signal to noise were selected, rotationally aligned and merged, with a total of 13,511 single images contributing to the construction of a final average diffraction pattern (Fig. 2c). Individual diffraction peaks visible within the X-ray diffraction data were identified and fitted with horizontal and vertical Gaussian distributions in order to generate a representation of the experimental data with all sources of noise between the diffraction spots removed (Fig. 2d).

**Iterative phase retrieval**. 2D projection images were generated by Fourier transform of each of the diffraction patterns shown in Fig. 2 using an iterative phase retrieval algorithm[24,38,39]. Initial phases were generated by assuming a featureless tube with inner and outer diameters of 17.4 and 25.4 nm, respectively (Fig. 4b). This model was also used to estimate the amplitude and shape of the central (forward scattering) peak, which is not possible to measure experimentally. 100 rounds of iterative cycles (Fig. 4a) were used to improve the phases. The key idea of this algorithm is that the target object is placed within a support that forces the electron density in projection to fall to zero outside of the object. This was implemented by multiplying the object by a smoothed top-hat function in the horizontal direction (amplitude = 1 for |x| ≤ 15.6 nm from the center of the tube and then falling to zero as a half Gaussian with width of 11.8 nm) and a Gaussian

function (full width half maximum = 159 nm) in the vertical direction that was maximal at the center of the image ($y = 0$). An algorithmic cycle, depicted in Fig. 4a, was implemented in MatLab which applied the above supporting mask; made a 2D Fourier transform of the modified object to recover candidate phases; and made an inverse Fourier transform of the experimental image using these phases to recover a new object. The algorithm's rate of convergence and sensitivity to the mask parameters were improved by applying an inversion symmetry about the center of the 2D object in real space, which was implemented every 10 cycles. After this image reconstruction procedure had converged, the object was divided by the Gaussian in the vertical direction so as to correct for the implementation of this mask, and the 2D projection density images that resulted from this procedure are shown in Fig. 5. The sensitivity of this algorithm on the initial starting model (the inner and outer diameter of the featureless tube, Fig. 4b) was tested by repeating this analysis but starting with a variety of initial inner and outer diameters, quantified as the radial COM (=sum(radius × mass within the radial shell)/total mass) chosen to cover the biologically relevant domain. The results from this analysis are presented in Supplementary Fig. 2. Various support function parameters were also explored and the results of iterative phase retrieval were not sensitive to these choices.

**Theoretical model to validate the phase retrieval algorithm**. A model of an oversimplified microtubule was made by arranging spheres of 4 nm in diameter as a helix with 14 spheres per turn and a pitch of 3 spheres per turn. This corresponded to an outer tube diameter of 26.4 nm. Projection images were generated from this model (Fig. 6) and were used as idealized data against which several iterations of the phase retrieval algorithm were applied, confirming the convergence of the algorithm.

## Data availibility

Data are available at the CXI database[58] (http://www.cxidb.org/) with identification number ID-92.

## Code availibility

In house MatLab scripts for iterative phase retrieval are available at the CXI database[58] under identification number ID-92.

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

## Acknowledgements

Use of the Linac Coherent Light Source (LCLS), SLAC National Accelerator Laboratory, is supported by the U.S. Department of Energy, Office of Science, Office of Basic Energy Sciences under Contract No. DE-AC02-76SF00515. Parts of the sample delivery system used at LCLS for this research was funded by the NIH Grant P41GM103393, formerly P41RR001209. We acknowledge SAXS beamtime at beamline I911-4 on the MAX II storage ring of the MAX IV Laboratory (formerly MAX-lab) and support by Dr Tomas Plivelic. We thank Fredrik Bråtner at Dalsjöfors Kött for providing us with calves' brains. We are grateful to Dr Gregory Stewart at SLAC National Accelerator Laboratory for help with preparing Fig. 1. R.N. acknowledges funding from the Knut and Alice Wallenberg Foundation (Grant Numbers KAW 2012.0284, KAW 2012.0275 and KAW 2014.0275), the Swedish Research Council (Grant Numbers 2015-00560 and 349-2011-6485) and the Swedish Foundation for Strategic Research (Grant Number SRL10-0036). We are grateful for support from the Helmholtz Association through project-oriented funds to DESY. R.C.R. was supported by A*STAR (Agency for Science, Technology and Research), Singapore. PHZ was supported by the National Institutes of Health (NIH) under Award R01GM109019.

## Author contributions

R.N., R.C.R. and H.N.C. conceived the experiment, which was designed with input from G.B., M.L. and S.B. Samples were prepared and characterized using TEM and SAXS by G. B. and R.H. The XFEL experiment was conducted by G.B., R.H., D.A., A.B., P.B., D.P.D., C.S., T.A.W., F.S., R.B., K.B., L.M.G.C., H.F., C.G., U.G., L.G., D.O., D.P., M.S., T.T., M. M., G.J.W., H.N.C., M.L., S.B., R.C.R. and R.N. with a particular focus on injection by D. P.D., F.S., R.B., K.B., L.M.G.C., H.F., C.G., L.G., D.O., T.T. and M.L. Data were processed by G.B., G.H., A.J., D.A., A.B., S.T. and T.A.W. and the iterative phase retrieval was performed by G.H. following discussions with G.B., E.M., N.D.L., P.Z. and R.N. The manuscript was prepared by R.N., G.B., and G.H. with additional input from all authors.
