## [Peer Review File · Nature Communications]

Reviewers' comments:

Reviewer #1 (Remarks to the Author):

Review of
COHERENT DIFFRACTIVE IMAGING OF MICROTUBULES USING AN X-RAY LASER
Authors: Brändén et.al.

This is a solid and technically sound article demonstrating a simple approach for recovering a 2D projection of microtubules by applying Electron Microscopy tools and a phase retrieval algorithm. This is a proof-of-principle that could be applied to any biological polymer exhibiting helical symmetry. Using a GDVN liquid microjet, they injected "aligned" microtubules across an XFEL beam. They collected millions of X-ray scattering images, thousands of images were selected and manually inspected for further processing. The final average diffraction pattern came from roughly 13,000 single images that were rotationally aligned and merged. This novel hybrid approach could have a huge impact to others in the community, and will help to study the dynamics of biological helical assemblies at room temperature.

I think this is a very important work and I hope the authors develop a semi-automatic workflow so other groups can use this approach. This article should be published after addressing these minor comments.

Minor Comments:

1) Is not clear from Figure 4C and 4D if the authors imposed symmetry to end with a 13-pf or 14-pf tubules.

Please confirm in the text that no additional symmetry was imposed.

2) The authors state that they see a 4 nm substructure in the 2D projections (Figure 4D) and from their analysis it is possible that they have a mixture of 13-pf and 14-pf. To strengthen your conclusions, the authors should perform a negative control. The same analysis but imposing the less likely variants 12-pf, 15-pf and 16-pf. Please state that the correlations are worse than the 13 and 14-pf.

3) From the description in the data processing and iterative phase retrieval sections, it seems that the process is largely driven by manual curation of the initial images. I would like to see a comment describing which step was really automatic and which one was manual and required intervention.

Reviewer #2 (Remarks to the Author):

The manuscript entitled "Coherent diffractive imaging of microtubules using an X-ray laser" by Brändén et al. describes a novel technique to analyze microtubule polymers via ultra-short X-ray pulses from an XFEL source.

X-ray free electron lasers (XFELs) represent the latest development in synchrotron technology. The ultra-high intensity and ultra-short X-ray pulses allow to investigate the structures of increasingly smaller and smaller samples. The properties of the X-ray source require the development of new analysis techniques with respect to sample preparation / sample mounting as well as data processing and analysis. The manuscript describes that latest advances to apply XFEL synchrotron radiation to a biological sample that is difficult to study by other means: microtubules.

In the current manuscript the authors describe a set of experiments that were conducted using brain-derived microtubules that were exposed to a highly focused XFEL X-ray beam. The samples were polymerized in vitro, diluted to allow spraying of the sample suspension, and exposed to

tightly focused X-ray pulses. The authors estimated that up to 20 microtubules were contained in the volume exposed to each X-ray pulse. The diffraction data were collected at a high frequency, analyzed in batches, and averaged according to image classifications. The resulting averages were used in an iterative phase retrieval algorithm to obtain real space 2D projections of the microtubule structure, which were able to discern the 4 nm tubulin lattice.

A notable weakness of the manuscript arises from the data processing as it is described. As it stands, it is not clear if it is a lack of clarity in the description, or if the processing procedure has some underlying flaws: the diffraction patterns are classified into five subgroups, which were then averaged. No information is provided how these subgroups were defined / selected. Given the orientational disorder that is inherent in the sample it is not clear how homogeneous the averaged subgroups truly were. Would it be possible to do a rotational alignment on the individual diffraction patterns before the classification is performed? Moreover, to what degree does the tiling of the diffraction pattern influence the classification? The hard edges of the tiling pattern could easily dominate the classification procedure and produce an artifact in the subgroup composition. It might be useful to do the classification on only one tile (the one closest to the diffraction origin). Lastly, why were the classifications limited to five subgroups? Since the microtubules in the sample could be disordered by varying degrees, it would make sense to use many more subgroups for classification purposes. In fact, many image processing procedures define the number of subgroups needed internally.

Additionally, the manuscript has a few other shortcomings that need to be addressed as well:

1. In the introduction (page 4) microtubule protofilaments are equated with 12 to 16 tubulin dimers "per turn". The helicity of the microtubules is dependent on the helical parameters, which don't directly reflect on the protofilament structure. The latter are usually straight, while most microtubule helices involve a "3-start" helix and a ~ 0.9 nm stagger between individual protofilaments. This sentence needs to be re-written to reflect the more complicated nature / structure of microtubules.
2. The cryo EM studies of microtubular structures also used "Zn-sheets" and electron crystallography approaches to obtain high-resolution structure information, not just single-particle techniques (introduction: page 4/5).
3. For the analysis / interpretation of the data the authors assume the density outside of the microtubules to be zero. Since the preparations contain "crude" microtubules, the density is certainly not zero. These brain-derived microtubules are densely decorated with microtubule-associated proteins e.g. MAP1, MAP2, tau, etc. While this information will not influence the relatively low-resolution analyses in this manuscript, it should be mentioned that there is quite a bit of (mostly disordered) density outside of these microtubules.
4. In the results section the authors mention that negative stain electron microscopy was used to confirm the quality of the microtubule preparations. It would be informative for the non-specialist reader to see a representative example of such an electron micrograph.
5. In the description of the microtubule formation (methods; page 13) the authors mention to have used purified α/β tubulin, but it is not mentioned how those purified α/β tubulins were obtained. The method described under "Protein preparation" would provide only "crude" microtubule preparations that are also enriched in microtubule-associated proteins such as MAP1, MAP2, and the tau protein. It would take several more steps to remove those MAPs from the tubulin. Please clarify the description in these paragraphs.

Reviewer's comments in blue, our response in black.

Reviewer #1 (Remarks to the Author):

Review of

COHERENT DIFFRACTIVE IMAGING OF MICROTUBULES USING AN X-RAY LASER

Authors: Brändén et.al.

This is a solid and technically sound article demonstrating a simple approach for recovering a 2D projection of microtubules by applying Electron Microscopy tools and a phase retrieval algorithm. This is a proof-of-principle that could be applied to any biological polymer exhibiting helical symmetry. Using a GDVN liquid microjet, they injected "aligned" microtubules across an XFEL beam. They collected millions of X-ray scattering images, thousands of images were selected and manually inspected for further processing. The final average diffraction pattern came from roughly 13,000 single images that were rotationally aligned and merged. This novel hybrid approach could have a huge impact to others in the community, and will help to study the dynamics of biological helical assemblies at room temperature.

I think this is a very important work and I hope the authors develop a semi-automatic workflow so other groups can use this approach. This article should be published after addressing these minor comments.

We thank Reviewer #1 for this accurate summary and positive recommendation.

Minor Comments:

1) Is not clear from Figure 4C and 4D if the authors imposed symmetry to end with a 13-pf or 14-pf tubules. Please confirm in the text that no additional symmetry was imposed.

With respect to Figure 4 (now Figure 5), we did not impose a 13-pf or 14-pf symmetry in the algorithm.

The phase retrieval algorithm did apply an inversion symmetry in the real-space representation of the image through the central point of the image. Inversion symmetry was applied in order to enhance the convergence of the algorithm. When this symmetry was relaxed the algorithm tended to oscillate about this solution, with one side and then the other side of the reconstructed projection image showing higher/lower projected electron density. We found that by imposing inversion symmetry every few iterations (in this case every ten iterative cycles) this oscillation was suppressed.

On page 16 of the revised manuscript we write:

"The algorithm's rate of convergence and sensitivity to the mask parameters were improved by applying an inversion symmetry about the centre of the 2D object in real space, which was implemented every 10 cycles."

2) The authors state that they see a 4 nm substructure in the 2D projections (Figure 4D) and from their analysis it is possible that they have a mixture of 13-pf and 14-pf. To strengthen your conclusions, the authors should perform a negative control. The same analysis but imposing the less likely variants 12-pf, 15-pf and 16-pf. Please state that the correlations are worse than the 13 and 14-pf.

As noted above, we did not impose 13-pf or 14-pf symmetry in our iterative phase retrieval algorithm. Therefore, the request to impose the symmetry of less likely variants such as 12-pf, 15-pf and 16-pf may be due to a slight misunderstanding. Nevertheless, we have addressed this request by providing more details of the results from the analysis when we used a variety of tube diameters as the starting point (now Figure 4B).

We now state (overlooked in the first submission) on page 9 that the physical inability to measure the forward scattered X-rays (the [0,0] peak in X-ray crystallography) means that this data is missing and therefore this has to be modelled. In our iterative phase retrieval algorithm we approximated the intensity and shape of the [0,0] peak from the starting featureless tube-model (Figure 4B). We then used the same approach to recover results from phase retrieval iterations using a set of starting unstructured tube models with a “centre-of-mass radius” from 9 nm to 16 nm, and these are shown in Supplementary Figure 2.

In Supplementary Figure 2A we plot the “centre-of-mass radius” for the starting model – the solid tubes (Figure 4B) – versus the peak-to-peak distance that results from the iterative phase retrieval algorithm. We see that over a number of starting radii from 10 nm to 15 nm they converge to a peak-to-peak separation of 19.9 ± 0.8 nm.

In Supplementary Figure 2B we plot the Pearson correlation function value, using the same “centre-of-mass radius” starting models for the initial phases, against the low-resolution peaks of the raw data (centre left inner panel of Figure 2C). This analysis gave a broad peak in the Pearson correlation function centred around 11 nm. When one approximates the 13-3 pf biological assembly of Nogales et al. (pdb entry 5SYF; Kellogg, E. H. et al. Insights into the Distinct Mechanisms of Action of Taxane and Non-Taxane Microtubule Stabilizers from Cryo-EM Structures. *J Mol Biol* **429**, 633 (2017)) as a featureless tube (Figure 4B) the corresponding value is 11.2 nm. The distribution in Supplementary Figure 2B is quite broad and therefore also consistent with a sample containing 14-pf microtubules.

Alternatively, by azimuthal-integration of the image shown in Figure 2C we can generate a pseudo SAXS image. This shows the first peak J_{01} in the SAXS pattern at 0.297 nm^{-1} . Using the formula J_{01} (in units of nm^{-1}) = $7.66/(2R)$ [Matesanz, R., et al. Modulation of microtubule interprotofilament interactions by modified taxanes. *Biophys J* **101**, 2970 (2011)] we get a value of R of 12.9 nm, which corresponds to an average protofilament number of around 14 according to Matesanz et al..

For these reasons we believe our sample corresponds to a mixture of 13-pf and 14-pf microtubules. These details have now been added to the manuscript on page 10 and 11.

3) From the description in the data processing and iterative phase retrieval sections, it seems that the process is largely driven by manual curation of the initial images. I would like to see a comment describing which step was really automatic and which one was manual and required intervention.

To address this request we have added a new Figure 3 which provides a detailed flow-chart of the steps involved in processing the data. The two points where manual interventions were applied are highlighted in this flow-chart.

Reviewer #2 (Remarks to the Author):

The manuscript entitled “Coherent diffractive imaging of microtubules using an X-ray laser” by Brändén et al. describes a novel technique to analyze microtubule polymers via ultra-short X-ray pulses from an XFEL source.

X-ray free electron lasers (XFELs) represent the latest development in synchrotron technology. The ultra-high intensity and ultra-short X-ray pulses allow to investigate the structures of increasingly smaller and smaller samples. The properties of the X-ray source require the development of new analysis techniques with respect to sample preparation / sample mounting as well as data processing and analysis. The manuscript describes that latest advances to apply XFEL synchrotron radiation to a biological sample that is difficult to study by other means: microtubules.

We thank Reviewer #2 for this accurate and positive summary.

In the current manuscript the authors describe a set of experiments that were conducted using brain-derived microtubules that were exposed to a highly focused XFEL X-ray beam. The samples were polymerized in vitro, diluted to allow spraying of the sample suspension, and exposed to tightly focused X-ray pulses. The authors estimated that up to 20 microtubules were contained in the volume exposed to each X-ray pulse. The diffraction data were collected at a high frequency, analyzed in batches, and averaged according to image classifications. The resulting averages were used in an iterative phase retrieval algorithm to obtain real space 2D projections of the microtubule structure, which were able to discern the 4 nm tubulin lattice.

This is an accurate summary of the steps involved.

A notable weakness of the manuscript arises from the data processing as it is described. As it stands, it is not clear if it is a lack of clarity in the description, or if the processing procedure has some underlying flaws:

In the revised manuscript we have included a flow-chart that explains the data-processing steps (Figure 3). We hope that this clarifies these issues.

the diffraction patterns are classified into five subgroups, which were then averaged. No information is provided how these subgroups were defined / selected.

The creation of sub-groups was automated using the program Xmipp. This programme was developed for single-particle electron microscopy applications and is an open-source image processing package. Details concerning how the algorithm works have been published by the authors of the programme [Sorzano, C. O. et al. XMIPP: a new generation of an open-source image processing package for electron microscopy. *J Struct Biol* **148**, 194-204 (2004)]. We now include this reference in addition to citing the practical applications paper [Scheres, S. H et al., Image processing for electron microscopy single-particle analysis using XMIPP. *Nat Protoc* **3**, 977-990 (2008)].

The classification algorithm is a self-organizing maps (SOMs) neural network algorithm. This SOMs approach is well accepted in the electron microscopy community and has been validated in numerous studies. We did not modify this algorithm. We did provide manual input by selecting the number of sub-groups, which we chose to be five sub-groups for sets of approximately 1000 images from any given run. This choice of five sub-groups was arbitrary, but provided a good compromise given the power of our computing system.

Given the orientational disorder that is inherent in the sample it is not clear how homogeneous the averaged subgroups truly were.

On average we estimate that data were collected from approximately 20 microtubules for each “useful” image from a single XFEL pulse. In an earlier publication we estimated that microfilaments (not microtubules) flow-aligned to an angular disorientation of five degrees within the same microjet [Popp, D. et al. Flow-aligned, single-shot fiber diffraction using a femtosecond X-ray free-electron laser. *Cytoskeleton* **74**, 472 (2017)]. We believe that the approach we use here of using XMIPP to sort into class averages provided a significant improvement on the angular orientation within the class averages (compare Figure 2B of this manuscript to Figures 2, 3 and 4 in Popp et al). Irrespectively, slight angular disorientation is not a limiting factor when it comes to the image reconstruction since data extend only to 2 nm resolution.

Would it be possible to do a rotational alignment on the individual diffraction patterns before the classification is performed?

This would be possible but it would not be useful given the current data. By using the sorting algorithm XMIPP the data is self-organized and classified into closely related images. This improves the signal-to-noise, and this improvement allows the class-average to be better rotationally aligned against other class-averages than when trying to do this with an individual image. In this manner we

have improved significantly upon the angular disorientation when simply averaging all data together (eg. Popp et al.).

Moreover, to what degree does the tiling of the diffraction pattern influence the classification? The hard edges of the tiling pattern could easily dominate the classification procedure and produce an artifact in the subgroup composition. It might be useful to do the classification on only one tile (the one closest to the diffraction origin).

The classification is dominated by two factors: the quality of the X-ray diffraction from the microtubules (some images show more peak-to-peak contrast and less background than others) and the instantaneous orientation of the microjet (and hence the microtubules) at the time of exposure to the XFEL beam. Thus Reviewer #2 is correct to suggest that the detector tiling is influencing the classification categories, since the orientation of the microjet may change from image to image and this changes the angle of the image relative to the detector (and its tiles). In this way class-averaging is compensating for fluctuations in the angle of the microjet during data collection.

We do not consider it necessary to repeat sorting analysis using only one tile with these data. In the future, if the microjet (or another) injection technology and XFEL focus and intensity allows individual microtubules to be exposed and useful data recorded, then the selection of individual detector tiles (versus the use of the entire detector) may be a useful parameter to explore when optimizing the sorting algorithm.

Lastly, why were the classifications limited to five subgroups? Since the microtubules in the sample could be disordered by varying degrees, it would make sense to use many more subgroups for classification purposed. In fact, many image processing procedures define the number of subgroups needed internally.

We made the decision to limit the sorting algorithm to five sub-groups on individual runs of approximately 1000 images per run. This was a pragmatic consideration based upon the computational cost of choosing more sub-groups and larger sub-sets of data versus the computing power available to us when working through the data. Because we worked with approximately 38 sets of 1000 images, then this led to 186 sub-groups in total (Figure 3). As the flow-chart shows (Figure 3), this set of 186 sub-groups containing 38 588 images in total, was further narrowed down to only 13 511 images due to signal-to-noise considerations, with only the sub-groups with the best signal to noise being kept.

Additionally, the manuscript has a few other shortcomings that need to be addressed as well:

1. In the introduction (page 4) microtubule protofilaments are equated with 12 to 16 tubulin dimers "per turn". The helicity of the microtubules is dependent on the helical parameters, which don't directly reflect on the protofilament structure. The latter are usually straight, while most microtubule helices involve a "3-start" helix and a ~0.9 nm stagger between individual protofilaments. This sentence needs to be re-written to reflect the more complicated nature / structure of microtubules.

This sentence has now been rewritten to read "Microtubules are built up of α/β tubulin dimers arranged as hollow cylinders with a mean diameter of approximately 24 nm and consist of 12 to 16 multiples of tubulin dimers per turn, the so-called proto-filaments, arranged in a staggered conformation with most helices arranged with a pitch of three tubulin monomers per turn."

2. The cryo EM studies of microtubular structures also used "Zn-sheets" and electron crystallography approaches to obtain high-resolution structure information, not just single-particle techniques (introduction: page 4/5).

This sentence has now been rewritten to read "Our current knowledge of the structure of microtubules therefore stems primarily from negative-stained electron microscopy (resolution down to ~3 nm⁶), X-ray fibre diffraction (resolution ~1 nm⁷), X-ray solution scattering (resolution ~3 nm^{8,9})

and cryo-electron microscopy studies¹⁰⁻¹⁵ which have achieved a resolution ($\sim 3.5 \text{ \AA}$) at which most side-chains can now be resolved.”

A new reference has been added (Nogales, E., Wolf, S. G., Khan, I. A., Luduena, R. F. & Downing, K. H. Structure of tubulin at 6.5 A and location of the taxol-binding site. *Nature* **375**, 424-427 (1995)) which used Zn induced crystalline sheets.

3. For the analysis / interpretation of the data the authors assume the density outside of the microtubules to be zero. Since the preparations contain "crude" microtubules, the density is certainly not zero. These brain-derived microtubules are densely decorated with microtubule-associated proteins e.g. MAP1, MAP2, tau, etc. While this information will not influence the relatively low-resolution analyses in this manuscript, it should be mentioned that there is quite a bit of (mostly disordered) density outside of these microtubules.

We followed the preparation of: M. Castoldia & A.V. Popov, Purification of brain tubulin through two cycles of polymerization–depolymerization in a high-molarity buffer, *Prot. Expr. Purif.* **32** 83 (2003) which produces MAP free tubulin. This can be seen from the gel below. The controls are purchased from Cytoskeleton Inc., whereas the in-house are our own preparations used in these experiments (in this case the bovine MTs). Mass spectrometry controls showed the main band to be tubulin. MAPs and tau etc... were not detected in any of these bands.

4. In the results section the authors mention that negative stain electron microscopy was used to confirm the quality of the microtubule preparations. It would be informative for the non-specialist reader to see a representative example of such an electron micrograph.

A representative image of a negative stain electron micrograph is now included as Supplementary Figure 1.

5. In the description of the microtubule formation (methods; page 13) the authors mention to have used purified α/β tubulin, but it is not mentioned how those purified α/β tubulins were obtained. The method described under "Protein preparation" would provide only "crude" microtubule preparations that are also enriched in microtubule-associated proteins such as MAP1, MAP2, and the tau protein. It would take several more steps to remove those MAPs from the tubulin. Please clarify the description in these paragraphs.

As described in response to point 3, the purification successfully removed MAPs etc... We feel that the citation above is sufficient to allow others to reproduce these preparations.

Reviewers' comments:

Reviewer #1 (Remarks to the Author):

Thank you for addressing all the questions and comments.

1) I am satisfied with the answers for points #1 and #2 in regards to imposing symmetry.

The clarification in pages 10 and 11 answers the questions I raised.

2) In regards to point #3, the authors improved the flow-chart (Figure #3) and now it is clear which steps will require manual intervention.

3) Since the authors have addressed all the points I consider that the article should be published.

4) One final issue to correct is that the Data and Scripts are not currently visible under identification XX at the CXIDB.

Reviewer #2 (Remarks to the Author):

The revised manuscript entitled "Coherent diffractive imaging of microtubules using an X-ray laser" by Brändén et al. is substantially improved over the initial submission.

All critiques have been addressed, and I have no further objections to its acceptance for publication in Nature Communications.

Reviewer #1 (Remarks to the Author):

Thank you for addressing all the questions and comments.

1) I am satisfied with the answers for points #1 and #2 in regards to imposing symmetry. The clarification in pages 10 and 11 answers the questions I raised.

2) In regards to point #3, the authors improved the flow-chart (Figure #3) and now it is clear which steps will require manual intervention.

3) Since the authors have addressed all the points I consider that the article should be published.

We thank Reviewer #1 for these clear answers to his/her previous points of concern.

4) One final issue to correct is that the Data and Scripts are not currently visible under identification XX at the CXIDB.

Both the experimental data and our in-house data analysis scripts have been made available on the CXI data base with accession code ID-92. This can be seen through this link:

<http://cxidb.org/id-92.html>

Reviewer #2 (Remarks to the Author):

The revised manuscript entitled “Coherent diffractive imaging of microtubules using an X-ray laser” by Brändén et al. is substantially improved over the initial submission.

All critiques have been addressed, and I have no further objections to its acceptance for publication in Nature Communications.

We thank Reviewer #2 for this clear answers to his/her previous points of concern.

REVIEWERS' COMMENTS:

Reviewer #1 (Remarks to the Author):

The authors have addressed the critique about the "data not shown" by adding a figure of the small-angle scattering of the microtubules. I consider that the article should be accepted for publication.